# Electroconvulsive shock and transcranial magnetic stimulation do not alter the survival or spine density of adult-born striatal neurons

Tara Gaertner[1,2], Tian Rui Zhang[1,2,3], Baran Askari[1,2,3], Fidel Vila-Rodriguez[1,2,4,5,6], Jason S. Snyder [2,3]*

1 Non-Invasive Neurostimulation Therapies Laboratory, Department of Psychiatry, Faculty of Medicine, University of British Columbia, Vancouver, British Columbia, Canada, 2 Djavad Mowafaghian Centre for Brain Health, University of British Columbia, Vancouver, British Columbia, Canada, 3 Department of Psychology, University of British Columbia, Vancouver, British Columbia, Canada, 4 School of Biomedical Engineering, University of British Columbia, Vancouver, British Columbia, Canada, 5 Spanish Consortium of Biomedical Research in Epidemiology and Public Health (CIBERESP), Madrid, Spain, 6 Health Technology Assessment in Primary Care and Mental Health Research Group (PRISMA), Barcelona, Spain

* jasonsnyder@psych.ubc.ca

## Abstract

Adult neurogenesis has most often been studied in the hippocampus and subventricular zone-olfactory bulb, where newborn neurons contribute to a variety of behaviors. A handful of studies have also investigated adult neurogenesis in other brain regions, but relatively little is known about the properties of neurons added to non-canonical areas. One such region is the striatum. Adult-born striatal neurons have been described in both rodents and humans, but the regulation of these neurons is poorly understood. Since striatal dysfunction occurs in Parkinson's disease, which is amenable to neurostimulation therapies, we investigated whether electroconvulsive shock (ECS) or transcranial magnetic stimulation (rTMS) modulate neuroplasticity of adult-born striatal neurons. Adult-born cells were labelled in transgenic mice and 8 days later mice were given 10 stimulations over the course of 3 weeks. Adult-born striatal neurons were consistently observed in all groups. Their dendritic morphology and expression of DARPP32 and NeuN indicated a medium spiny neuron phenotype. However, neither ECS nor rTMS altered the number of new neurons, and both treatments also had no effect on the density of dendritic spines compared to unstimulated controls. These results suggest that neither ECS nor rTMS alter early neuronal survival or morphological plasticity at postsynaptic sites in the striatum.

## Introduction

Non-invasive forms of neurostimulation such as electroconvulsive therapy (ECT) and repetitive transcranial magnetic stimulation (rTMS) are used to treat an array of neuropsychiatric disorders [1–3] and, while the mechanisms are not fully elucidated, it is hypothesized that some of their therapeutic effects might depend on neuroplasticity. Specifically, rTMS and ECT are commonly prescribed for mood disorders such as depression. In humans, both treatments

**Data availability statement:** All data are provided as Supporting Information.

**Funding:** FVR receives research support from CIHR, Brain Canada, Michael Smith Foundation for Health Research, Vancouver Coastal Health Research Institute, and Weston Brain Institute for investigator-initiated research. FVR receives philanthropic support from Seedlings Foundation. FVR received in-kind equipment support for this investigator-initiated trial from MagVenture. JSS receives research support from CIHR and NSERC. The funders had no role in study design, data collection and analysis, decision to publish, or preparation of the manuscript."

**Competing interests:** FVR has received honoraria for participation in an advisory board for Allergan. FVR is a volunteer director on the board of directors of the British Columbia Schizophrenia Society. This does not alter our adherence to PLOS ONE policies on sharing data and materials.

lead to plastic changes in the prefrontal cortex that may contribute to therapeutic effects [4]. rTMS and electroconvulsive shock (ECS; the animal model of ECT) also induce profound plasticity in the hippocampus of rodents. ECS robustly increases both the birth [5,6] and survival (Zhang et al., unpublished observations) of newborn hippocampal neurons, and neurogenesis is necessary for the effects of ECS on depression-like behavior in mice [7]. While less is known about the neurobiological effects of rTMS, we have recently found that it promotes growth of presynaptic terminals of adult-born hippocampal neurons in male mice [8], and there is evidence (albeit mixed) that it increases the production of adult-born hippocampal neurons [8–11].

Given the broad biological basis of most psychiatric disorders, and strong connectivity between brain regions that may be targeted by ECT and rTMS [12,13], investigations into neurostimulation-induced plasticity in other brain regions is also warranted. For example, depression is often characterized by altered reward-based decision-making, reduced motivation and psychomotor disturbances that implicate the dorsal and ventral striatum [14,15]. ECT is also used to treat other disorders such as schizophrenia and Parkinson's disease, both of which are associated with dysfunction or degeneration in striatal circuitry [2,3]. Indeed, there is evidence that ECS and rTMS exert plastic changes in the striatum in animal models [16], including increasing neurogenesis [17]. While not typically appreciated as a site of adult neurogenesis, there are many reports of neurogenic plasticity in both the dorsal and ventral striatum. Newborn striatal neurons arise from the subventricular zone and migrate and mature into neuronal phenotype(s) characterized by expression of calretinin, NeuN and DARPP32 [18–21]. Early work demonstrated that striatal neurogenesis can be upregulated following stroke, contributing to the replacement of lost neurons [22,23]. More recent studies indicate that neurogenesis in the ventral striatum is increased in response to chronic pain [24]. Thus, striatal neurogenesis is plastic in response to pathology but the extent to which it is plastic in response to relevant forms of neurostimulation remains unknown. And while ECS may induce proliferation of new calretinin-positive interneurons [17], it is unknown whether ECS similarly promotes the survival of new neurons or alters their morphology. To address these questions, here we used transgenic mice to label adult-born neurons and examine whether ECT and rTMS alter the survival and spine density of immature neurons in the striatum.

## Methods

### Animals

Animal procedures were conducted as described recently by our group [8,25]. Tissue from the rTMS experiments were from the same animals used in Zhang et al. (2023) [8]. All procedures were approved by the Animal Care Committee at the University of British Columbia and conducted in accordance with the Canadian Council on Animal Care guidelines. Ascl1CreERT2 mice [26,27] were bred with CAGfloxStopTdTomato mice (Ai14 [28]) to generate experimental offspring where adult-born neurons could be labelled via tamoxifen injection. A total of 29 male and female mice were used (ECS, n = 6; sham ECS, n = 7; iTBS n = 8; sham iTBS, n = 8). There were equal numbers of males and females in each group, with the exception of the ECS sham group having 3 male mice and 4 female mice. Sexes were balanced and did not differ between groups. Mice were between 6-16 weeks of age at the time of tamoxifen injection (see below) and ages were similarly balanced across groups. Mice had ad libitum access to food and water and were housed on a 12 h light/dark cycle.

Animals were handled starting from 2 weeks prior to tamoxifen (TAM) injection and were briefly manually restrained for 1 min per day for 2 days prior to injection to reduce stress during intraperitoneal injections. TAM (25 mg/kg in sunflower oil) was injected 8

days prior to rTMS stimulation to induce tdTomato expression in newborn neurons (see Fig 1a for timeline).

## Neurostimulation

Neurostimulation was conducted as in Zhang et al [25] (described briefly below) except that the animals received sham or iTBS rTMS stimulation, or sham or ECS stimulation, every 2 days for a total of 10 sessions over 20 days. The animals were anesthetized with isoflurane, perfused with 4% paraformaldehyde, and their brains were collected 3 days following the last stimulation session.

Bilateral auricular ECS was administered to isoflurane-anesthetized mice via padded ear clip electrodes connected to a pulse generator (ECT Unit; Ugo Basile, Gemonio, Italy). Stimulation parameters were adapted from Yanpallewar et al. [29] and set at 0.5 ms pulse width, 50 Hz frequency, 0.5 s total shock duration, and 20–24 mA current. Sham animals were handled and ear clipped in the same fashion (for 1 min) but did not receive any stimulation. Stimulated animals were removed and placed back to the home cage once they regained posture and motor function.

iTBS was administered using a MagPro X100 device and a rTMS coil adapted for rodents (Cool-40 coil [30]; both from Mag Venture, Farum, Denmark). The average motor threshold was determined with a separate group of animals, before the main experiment, by stimulation with increasing power until a hindlimb motor response was evoked. Experimental

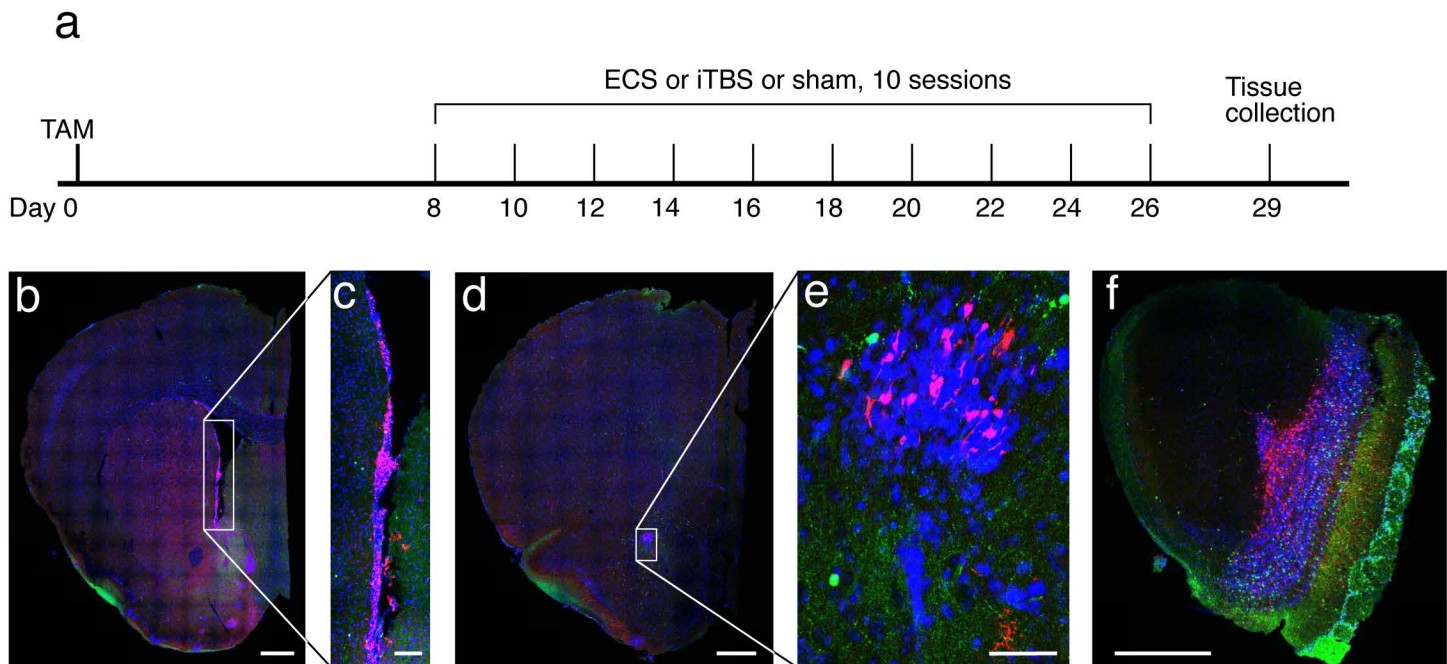

**Fig 1. tdTomato + cells in the SVZ, RMS and olfactory bulb in transgenic mice treated with tamoxifen.** (a) Timeline of experimental treatment. On day 0, mice were injected with tamoxifen to induce tdTomato expression. On day 8 and alternate days thereafter for 10 sessions, mice were given sessions of sham stimulation, ECS, or iTBS. Mice were euthanized on day 29. (b) to (f) Representative confocal images of coronal slices stained for tdTomato (red), calretinin (green) and DAPI (blue). (b) Hemisection through the striatum. Scale bar = 500 μm. (c) Enlargement of boxed region from b, showing tdTomato-expressing cells in the SVZ. Scale bar = 100 μm. (d) Hemisection through the anterior tip of the nucleus accumbens. Scale bar = 500 μm. (e) Enlargement of boxed region from d showing tdTomato-expressing cells in the rostral migratory stream. Scale bar = 50 μm. (f) Coronal section of olfactory bulb and olfactory nucleus showing TD-tomato-expressing cells migrating away from the rostral migratory stream into the granule layer of the olfactory bulb. Scale bar = 500 μm.

stimulation power was then defined as 115% of the motor threshold, which was 32% maximal stimulus output. Based on previous work [30], the di/dt follows a linear response and is 64A/μsec at 32%. Stimulation used a theta-burst protocol: triplet 50 Hz bursts were repeated at 5 Hz, with a 2 s train duration and an 8 s inter–train interval over 20 trains, for a total of 600 pulses over 3 min 9 s. Sham-stimulated mice were restrained for an equivalent amount of time and exposed to the rTMS coil but the stimulation intensity was set to 0%. While rTMS is often delivered daily in the clinic, here we opted to stimulate every other day to keep the schedule aligned with the ECS schedule. Furthermore, there is evidence that rTMS can be delivered every other day and still be as efficacious as daily treatment in the clinic [31].

## Tissue preparation and Immunohistochemistry

Brain tissue was processed as described [8]. Briefly, brains were fixed in paraformaldehyde, bisected, sliced coronally, and stored in cryoprotectant at −20°C until immunohistochemical processing. For free floating fluorescent double immunostaining, 40 μm hemi-brain sections were retrieved from the cryoprotectant solution and washed in 24-well plates with PBS. The slices were then immersed in 5% horse serum in 0.5% PBS-Triton for 30 min to block non-specific binding of the antibodies. The tissue was then incubated with primary monoclonal antibodies to either calretinin, DARPP-32 or NeuN plus rabbit anti-RFP antibody (1:2000; Sigma-Aldrich) at 4°C. Mouse anti-calretinin (Swant, 6B3) was used at 1:2000 and incubated 1-2 days at 4°C. Anti-DARPP (Santa Cruz Biotechnology, sc271111) was used at 1:250 and incubated 3 days at 4°C. Anti-NeuN (Millipore, MAB377) was used at 1:400 and incubated 3 days at 4°C. After the primary antibody incubation, slices were washed and then incubated and stained with their respective secondary antibodies for 1 h (1:250 donkey anti-mouse Alexa 488 antibody or 1:250 donkey-anti-mouse Alexa 647 antibody; 1:250 donkey antirabbit Alexa 555 antibody, all secondary antibodies from Sigma-Aldrich). All slices were stained with 1:1000 DAPI to visualize cell nuclei. Brain slices were mounted on coated slides and cover-slipped with PVA-DABCO to prevent fluorescent fading.

## Imaging and quantification

Images of the subventricular zone (SVZ), rostral migratory stream (RMS), and RFP-expressing striatal neurons were acquired at a size of 1024 × 1024 at a z-resolution of 1.5 μm.

For cell counts, RFP-labeled cells in the dorsal and ventral striatum were manually counted on a Leica fluorescence microscope with a 25× objective (numerical aperture 0.95). A 1 in 6 series of sections, 4–6 slices per mouse, was spanning images 39–65 in the Allen Mouse Brain Atlas [32]. Striatal areas were measured from images that were acquired with a Leica SP8 confocal microscope. Images were acquired with a 25X water immersion lens at 1.0 zoom (numerical aperture 0.95). Images of 64 × 64 pixels (400 Hz scan speed) in size were merged to encompass the entire brain section and used for measuring striatal volumes. Tissue volume was obtained by tracing the two-dimensional area of the striatum using ImageJ and multiplying the area by the tissue thickness. Positive cell densities (in $mm^3$) were then obtained by dividing cell counts by the total volume.

For dendritic protrusions, images were acquired with a glycerol-immersion 63X objective (numerical aperture 1.3) at 3X zoom, at a z-resolution of 0.5 μm. Dendritic protrusions/spines were counted manually on ImageJ with Cell Counter and normalized to the length of the dendritic segment. Protrusions on two to three dendritic segments were counted, and averaged, for each tdTomato-expressing cell. Between 1–6 cells, spanning the dorsal and ventral striatum, were analyzed per mouse.

## Statistical analysis

All underlying data are provided as Supporting Information (S1 File.xlsx). Effects of stimulation on tdTomato + cell density were assessed with unpaired t-tests. Effects of stimulation on spine density were assessed by linear mixed effects models in R (nlme package), to account for correlations between cells that are sampled from the same mouse [33]. In all cases, alpha was set at 0.05

## Results

### Newborn cells are present in the subventricular zone, rostral migratory stream and granule layer of the olfactory bulb

Using a fluorescence microscope, we broadly inspected coronal slices for the presence of newborn tdTomato-expressing cells. Although the presence of tdTomato + cells varied, tdTomato-expressing cells were consistently visible throughout the SVZ and the RMS (Fig 1b-1e), consistent with tamoxifen-induced expression in neuronal precursor cells in these regions [26]. At the anterior end of the RMS, tdTomato-expressing cells appeared to be migrating away from the RMS to take up their places in the granule layer of the olfactory bulb (Fig 1f). Similarly, at the ventral tip of the SVZ, cells appeared to migrate in both medial and lateral directions (Fig 2). Although previous studies have suggested that newborn neurons originating the SVZ may co-express calretinin [17,18,21], we found that co-expression of calretinin in tdTomato-expressing cells was limited to a small minority of cells in the granule layer, with no co-expression seen in the SVZ or RMS.

### Newborn (tdTomato-expressing) neurons are present in the striatum and their density is not altered by ECS or iTBS

On visual inspection of areas outside the SVZ and RMS, we noted that occasional tdTomato-expressing cells were present in a number of areas of the brain. In particular, cells with glial and oligodendrocytic morphologies were seen in the corpus callosum and anterior commissure, and cells with neuronal morphology were seen in the dorsal and ventral areas of the striatum. The striatal tdTomato-expressing cells had large, round cell bodies and elaborate trees of dendrites bearing spines, consistent with the morphology of medium spinal neurons (Fig 2b–2e). These cells were not exceedingly common but they were consistently found in each mouse (0–3 cells per hemisection). Cells were found throughout the striatum, both in the caudo-putamen and in the nucleus accumbens. Their identity as medium spiny neurons was confirmed by double-labelling for tdTomato and either the medium spiny neuron marker DARPP-32 (Fig 2k–2o) or the pan-neuronal marker NeuN (Fig 2q–2u).

Since ECS [5,6] has been previously found to increase adult neurogenesis in the hippocampus, and there is evidence that ECS increases the birth of new striatal neurons [17], we hypothesized that neurostimulation might also increase the survival of neurons that were born shortly before treatment. However, the density of tdTomato-expressing neurons in the striatum did not differ between mice who were administered 10 sessions of iTBS or ECS and those that received sham-stimulation (iTBS treated vs sham, $T_{14} = 0.1$; P = 0.92; ECS treated vs sham, $T_{11} = 0.7$, P = 0.51; Fig 2v). While we did not have sufficient group sizes to investigate sex differences, it is worth noting that ECS-treated males had more than twice as many new neurons than their sham counterparts, but the two groups of females were virtually identical (mean values: male sham, 2.2 cells/mm³; male ECS 5.2 cells/mm³; female sham, 5.9 cells/mm³; female ECS, 5.7 cells/mm³).

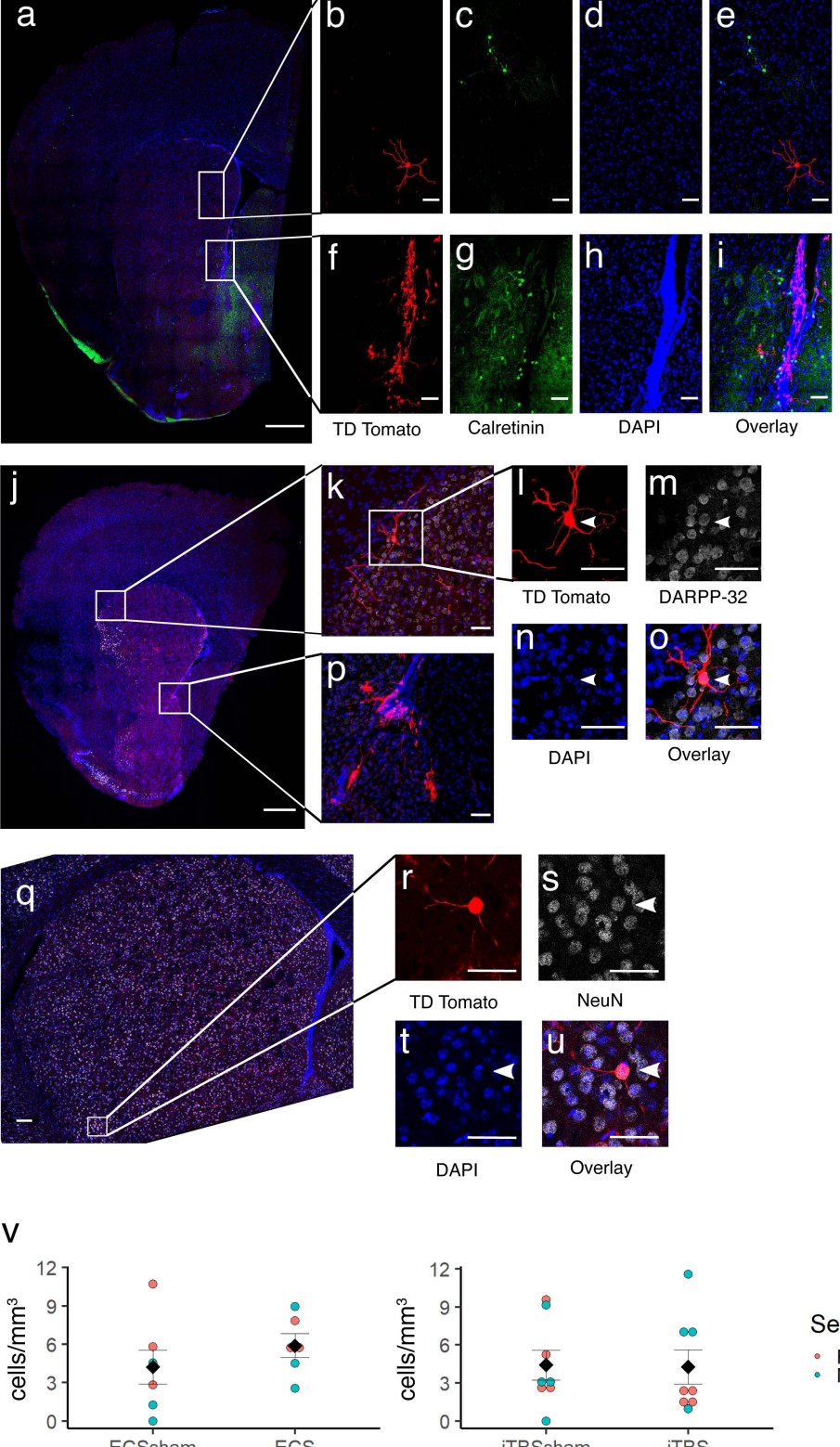

**Fig 2. tdTomato-expressing cells in the striatum express NeuN and DARPP-32 but not calretinin.** (a) Coronal hemisection through the striatum stained for tdTomato (red), calretinin (green) and DAPI (blue). Scale bar = 500 μm. (b)-(e) Enlarged insets from showing a tdTomato-expressing MSN which does not co-localize with calretinin. Scale bars = 50 μm. (f)-(i) Enlarged insets showing the ventral end of the SVZ where there is lack of colocalization

of tdTomato and calretinin staining. Scale bars = 50 µm. (j) Coronal hemisection through the striatum stained for tdTomato (red), DARPP-32 (white) and DAPI (blue). Scale bar = 500 µm. (k) Enlarged inset showing tdTomato-expressing MSN at dorsolateral edge of striatum. Scale bar = 50 µm. (l)-(o) Enlargement of neuron shown in k (arrowhead), confirming that tdTomato-expressing neurons express DARPP-32. Scale bars = 50 µm. (p) Enlarged inset showing tdTomato-expressing cells which appear to be migrating away from the ventral SVZ. Scale bar = 50 µm. (q) Coronal section of striatum stained for tdTomato (red), NeuN (white), and DAPI (blue). Scale bar = 100 µm. (r)-(u) Enlargement of boxed region from q showing a tdTomato-expressing cell (arrowhead) that also expresses NeuN. Scale bars = 50 µm. (v) Density of tdTomato-expressing cells in the striatum, by treatment group. Each colored circle represents data from an individual mouse. Black diamonds represent mean +/- standard error.

## The density of dendritic spines on newborn striatal neurons is not altered by ECS or iTBS

As noted above, striatal neurons expressing tdTomato were seen to have elaborate dendritic trees with prominent spines. To determine whether neurostimulation induced structural plasticity in adult-born striatal neurons, we quantified the density of dendritic spines in stimulated and sham-stimulated groups. Here, we found no difference in dendritic spine density on adult-born neurons of mice who had received ECS or iTBS compared to sham-stimulated mice (iTBS treated vs sham, P = 0.64, n = 27 cells from 14 mice; ECS treated vs sham, P = 0.94, n = 35 cells from 12 mice; Fig 3).

## Discussion

Non-invasive neurostimulation techniques have widespread effects on neural plasticity and cellular physiology, which may contribute to their effects on behavior [34,35]. Both are well known to promote neurogenic plasticity in the hippocampus [5,8,10,25] but far less is known about their effects on adult neurogenesis in other brain regions. To our knowledge, no study has examined how both ECS and iTBS impact striatal neurogenesis and the morphological properties of immature striatal neurons. Here, we therefore investigated whether ECS and iTBS promote the survival and morphological plasticity of adult-born neurons in the striatum, a brain region that is not typically studied in the context of adult neurogenesis. Newborn medium spiny neurons were observed in all groups of mice, and neither the numbers of new cells nor the density of their dendritic spines was altered by neurostimulation.

Our findings would seem to conflict with a previous report by Inta et al. that observed dramatic elevations in striatal neurogenesis following chronic ECS [17]. However, there are several notable differences between the 2 studies. First, by injecting BrdU during the window of ECS treatment, Inta et al. labelled neurons born *during* ECS (i.e., a proliferation effect). In contrast, we injected tamoxifen 8 days prior to the onset of ECS to test for changes in the early survival of new neurons (but not their proliferation). Thus, the 2 studies were designed to study different populations of cells. Differences in morphology and marker expression suggest that these were in fact distinct classes of cells altogether. ECS-induced proliferation resulted in the generation of small, calretinin-positive interneurons. In contrast, the tdTomato+ neurons we observed were large, multipolar, calretinin-negative and DARPP-32-positive, consistent with a medium spiny projection neuron phenotype. That adult-born striatal calretinin-negative interneurons have been identified in rabbits [36], rats [17,18] and humans [21], but not mice, has led to speculation that there may be species differences in the identity of adult-born striatal neurons [37]. In contrast, adult-born DARPP-32-positive medium spiny neurons have been identified in mice (here and [24]) and rats [22,23], but not humans [21]. Work in rats has identified newborn dorsal striatum medium spiny neurons mainly after stroke [22,23], but recent work in mice has observed medium spiny neurons even in the healthy brain, but they were restricted to the ventral striatum [24]. Here, we found medium

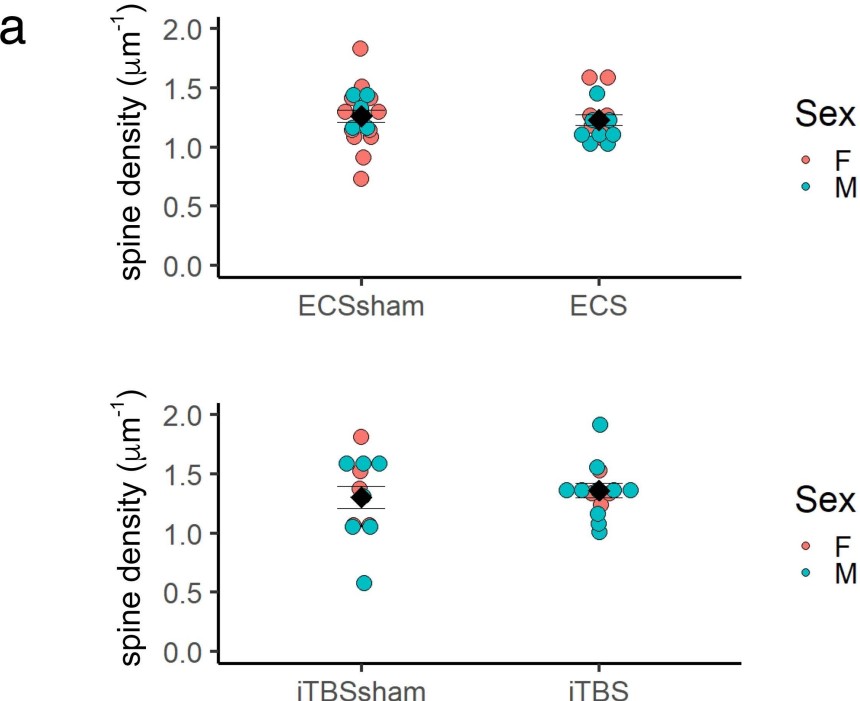

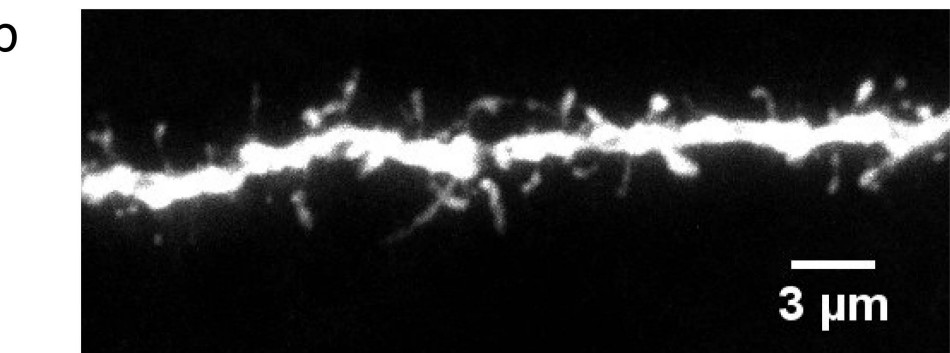

**Fig 3. Density of dendritic spines on tdTomato-expressing MSNs are not affected by neurostimulation treatments.**
(a) Density of dendritic spines on tdTomato-expressing neurons in the striatum, by treatment group. Each colored circle represents data from an individual neuron. Black diamonds represent mean +/− standard error. (b) Maximum-intensity projection of a section of dendrite from a tdTomato-expression in the striatum, showing dendritic spines.

spiny neurons in both the dorsal and ventral striatum of mice independent of pathology or neurostimulation treatment. Given these mixed results, future work may focus on the species, types of neurons, subregions and conditions that are required for neurogenesis in the adult striatum.

An outstanding question is whether striatal neurogenesis has any functional consequences. Here, new neurons were infrequent compared to those typically observed in the SVZ or hippocampus. However, mice only received a single injection of tamoxifen and the striatal labelling efficiency in AsclCreER mice is unknown, making it difficult to estimate the magnitude

of neurogenesis. Due to small numbers of cells present, spine densities were quantified per cell rather than by calculating animal-level averages. While it will be important for future studies to increase the within-animal sampling rate, by using mixed-effects models we were able to account for any animal-level variation in our analyses [33]. With respect to the potentially low rates of neurogenesis, it is worth noting that in the hippocampus unique physiological properties can endow even small numbers of new neurons with potent circuit-level and behavioral effects [38]. Low rates of neurogenesis, over extended portions of the lifespan, can also produce a substantial fraction of neurons [39]. Thus, it will be important for future studies to determine the functional relevance of adult striatal neurogenesis.

We focussed on the striatum not only because relatively little is known about its neurogenic capacity, but also because of its behavioral significance. The dorsal striatum is classically associated with motor behavior and reinforcement learning and the ventral striatum with motivating goal and reward-directed behaviors [40]. The striatum has therefore become a target of ECT and rTMS, particularly within the context of Parkinson's disease. There is evidence that, compared to rTMS, ECT induces greater structural plasticity (enlargement) in the striatum [41]. Our findings suggest that the survival of immature, pre-existing neurons is not involved in any large-scale structural changes.

Our negative results do not rule out the possibility that striatal neurogenesis might be modified by ECT and rTMS and/or exert a meaningful impact. As discussed above, stimulation could promote the birth of new cells and these cells could also have a physiological impact that is not captured by morphological measures such as spine density. It is also possible that other patterns of rTMS or ECS stimulation might induce neurogenic plasticity, possibly by altering other aspects of neuronal morphology such as dendritic complexity [42]. One limitation of our TMS coil is that it cannot provide the same level of regional stimulation in mice that is observed in human TMS. Thus it is likely that large regions of the mouse brain were stimulated. However, it seems unlikely that relatively widespread stimulation could explain the lack of effects on cell survival and spine plasticity, especially since we observed morphological plasticity in the hippocamus in the same mice [8]. Notably, ECS-treated male mice had more than twice as many newborn neurons than sham-treated male mice. While sample sizes were too low to draw any conclusions about sex differences, this may be a topic for future studies given that Parkinson's disease is more prevalent in men than in women [43]. While we did not observe any differences in structural plasticity, little is known about the developmental timecourse of newborn striatal neurons and so it is also possible that neurostimulation might induce plasticity at earlier or later timpoints than we investigated. Another consideration is the fact that we did not use a disease model, and so it remains possible that the production and properties of newborn striatal neurons might change in response to pathology, opening up the potential for therapeutic effects that are not possible in healthy animals.

## Supporting Information

**S1 File**. **Underlying data.** The complete dataset for all analyses is provided as S1_File.xlsx. (XLSX)

## Author contributions

**Conceptualization:** Tara Gaertner, Tian Rui Zhang, Fidel Vila-Rodriguez, Jason S. Snyder.

**Formal analysis:** Tara Gaertner, Jason S. Snyder.

**Funding acquisition:** Fidel Vila-Rodriguez, Jason S. Snyder.

**Investigation:** Tara Gaertner, Baran Askari.

**Methodology:** Tara Gaertner, Tian Rui Zhang, Baran Askari.

**Resources:** Fidel Vila-Rodriguez.

**Supervision:** Fidel Vila-Rodriguez, Jason S. Snyder.

**Writing – original draft:** Tara Gaertner.

**Writing – review & editing:** Fidel Vila-Rodriguez, Jason S. Snyder.

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
