## [Decision Letter · Decision Letter 0]

23 Sep 2024

PONE-D-24-35662Electroconvulsive shock and transcranial magnetic stimulation do not alter the survival or spine density of adult-born striatal neuronsPLOS ONE

Dear Dr. Snyder, Thank you for submitting your manuscript to PLOS ONE. After careful consideration, we feel that it has merit but does not fully meet PLOS ONE’s publication criteria as it currently stands. Therefore, we invite you to submit a revised version of the manuscript that addresses the points raised during the review process. 

After careful consideration by 3 Reviewers and an Academic Editor, all of the critiques of the Reviewers must be addressed in detail in a revision to determine publication status. If you are prepared to undertake the work required, I would be pleased to reconsider my decision, but revision of the original submission without directly addressing the critiques of the Reviewers does not guarantee acceptance for publication in PLOS ONE. If the authors do not feel that the queries can be addressed, please consider submitting to another publication medium. A revised submission will be sent out for re-review. The authors are urged to have the manuscript given a hard copyedit for syntax and grammar.

Please submit your revised manuscript by Nov 07 2024 11:59PM. If you will need more time than this to complete your revisions, please reply to this message or contact the journal office at plosone@plos.org . Please include the following items when submitting your revised manuscript:

We look forward to receiving your revised manuscript.

Kind regards,

Stephen D. Ginsberg, Ph.D.

Section Editor

PLOS ONE

Journal Requirements:

“FVR receives research support from CIHR, Brain Canada, Michael Smith Foundation for Health Research, Vancouver Coastal Health Research Institute, and Weston Brain Institute for investigator-initiated research. FVR receives philanthropic support from Seedlings Foundation. FVR received in-kind equipment support for this investigator-initiated trial from MagVenture.”

“I have read the journal's policy and the authors of this manuscript have the following competing interests:FVR has received honoraria for participation in an advisory board for Allergan. FVR is a volunteer director on the board of directors of the British Columbia Schizophrenia Society.”

5. We note that your Data Availability Statement is currently as follows: [All data will be provided as Supporting Information.]

Please confirm at this time whether or not your submission contains all raw data required to replicate the results of your study. Authors must share the “minimal data set” for their submission. PLOS defines the minimal data set to consist of the data required to replicate all study findings reported in the article, as well as related metadata and methods (https://journals.plos.org/plosone/s/data-availability#loc-minimal-data-set-definition ).

If your submission does not contain these data, please either upload them as Supporting Information files or deposit them to a stable, public repository and provide us with the relevant URLs, DOIs, or accession numbers. For a list of recommended repositories, please see https://journals.plos.org/plosone/s/recommended-repositories .

6. We note that you have referenced (unpublished observations) on page 2, which has currently not yet been accepted for publication. Please remove this from your References and amend this to state in the body of your manuscript: (ie “Bewick et al. [Unpublished]”) as detailed online in our guide for authors

7. Your ethics statement should only appear in the Methods section of your manuscript. If your ethics statement is written in any section besides the Methods, please delete it from any other section.

Reviewer's Comment:

**Comments to the Author**

1. Is the manuscript technically sound, and do the data support the conclusions?

Reviewer #1: Yes

Reviewer #2: Yes

Reviewer #3: Yes

2. Has the statistical analysis been performed appropriately and rigorously? 

Reviewer #1: Yes

Reviewer #2: Yes

Reviewer #3: Yes

3. Have the authors made all data underlying the findings in their manuscript fully available?

Reviewer #1: Yes

Reviewer #2: Yes

Reviewer #3: Yes

4. Is the manuscript presented in an intelligible fashion and written in standard English?

Reviewer #1: Yes

Reviewer #2: No

Reviewer #3: Yes

5. Review Comments to the Author

Reviewer #1: Gaertner et al have investigated the effect of two neurostimulation techniques on the survival or adult born striatal neurons. The study specifically studies the effect of the stimulation on neurons born prior to the stimulation regime. Although the study has some small sample sizes with some animals only have 1 neuron counted for dendritic spine density, it is well designed and well written. Publishing negative results is critical and informative and the authors should be congratulated on publishing these negative findings.

I only have minor corrections that should be easy to address.

Methods:

- please provide an estimate of the electric field induced or at least a dB/dT value for the rTMS stimulation. Presumably a biphasic sine wave was used?

- please check whether the duration listed (189 seconds) for iTBS is correct.

- Was there a rationale for delivering rTMS every second day? rTMS is most commonly given daily for a period. An alternative protocol is fine to use, but a description of how that rationale was chosen would be useful.

- please provide the NA for the objectives used.

- was there a reason 1-6 cells was counted per animal for dendritic spines? Counting 1 neuron vs 6 neurons is likely to have a large impact on the interpretation of the results.

Discussion:

- It would be helpful to discuss the impact of the following on the results and their interpretation.

(1) the use of an rTMS coil that stimulates the entire brain and at least some of the spinal cord if not most of the peripheral nerves. This coil is known to activate neurons in the spinal cord of rats, and would therefore be reasonable to assume is not focal at all for the mouse brain.

(2) the use of a non-standard rTMS protocol. Potentially a lack of effect could be due to the spacing of the stimulation. While the results are interesting, noting that other rTMS protocols could still have potential is worth noting.

Reviewer #2: The ms by Gaertner and colleagues describes the effects of ECT and rTMS on the survival and spine density of adult-born striatal neurons in mice. Both techniques did not alter striatal neurons survival and structural plasticity with the adopted stimulation patterns, which still needs to be pointed out.

Though of interest, the ms suffers from an unclear vocabulary, thus needing a strong language revision.

I have some major and few minor concerns.

Major points:

Introduction: Recent literature of rTMS effects on hippocampus, mPFC and M1 structural plasticity reported a stimulation-dependent modulation in mice; high-frequency rTMS increased dendritic plasticity in prefrontal and motor cortex, while 1 Hz stimulation induced structural plasticity in mature granule cells, as also dendritic complexity of newly generated neurons, though not increasing DG progenitor cells proliferation. These data shuld be discussed.

Methods: animals should be be better described in the initial paragraph.

Methods: Why did the average motor threshold was determined with a separate group of animals?

Results: better to specify what the markers RFP, DARPP-32 and NeuN are used for.

Minor points:

Introduction: The sentence "rTMS and electroconvulsive shock (ECS), the animal model of ECT,..." is not clear at all. Please rephrase.

Introduction: The sentence "More recent studies indicate that neurogenesis in the ventral striatum is increased in response to chronic pain" needs more details.

Methods: The sentence "Sham-stimulated mice were restrained and treated identically with the exception that the stimulation intensity" is not at all clear!! Please rephrase.

Methods: specify the microscope and objective used for the acquisition of images

Figure legend 3: modify "Each colored circle" with "each dot".

Reviewer #3: In their manuscript, Gaertner et al have investigated the effects of electroconvulsive shock (ECT) and 10Hz repetitive transcranial magnetic stimulation (rTMS) on the survival of adult born striatal neurons in the mouse brain. The author’s have used a transgenic mouse line to label adult-born neurons via tamoxifen injection, and beginning from 8 days later, applied ECT or iTBS rTMS for 10 sessions over 3 weeks. Neither ECT or rTMS induced changes in the amount of surviving adult-born striatal neurons or the density of dendritic spines. The manuscript is written well the findings are of interest to the field. My comments are detailed below:

Major comments

-The authors describe that they administered rTMS using a Cool-40 coil at 115% of motor threshold. Can the authors please provide detail on the estimated magnetic field or electric field at the surface of the cortex in their model. Additionally, there is evidence to suggest that this coil may lack focality and stimulate regions of the brain and spinal cord outside of the targeted region (https://doi.org/10.1111/ner.12387 ). Some acknowledgements of the non-focality of this coil should be included in the methods or discussion.

-Were animals that were delivered ECT anaesthetised for the duration of the stimulation or was this conducted in non-anesthetised animals? If it was conducted in anesthetised animals, can details of the anaesthesia protocol please be detailed.

-Can the authors please provide information on why the stimulation protocol was chosen. Why was stimulation delivered every second day? Also, why was tissue collected 3 days after the final stimulation day? Some discussion of how this stimulation protocol compares to other stimulation protocols and how the timing of tissue processing may have contributed to the results should be included.

-The discussion does not discuss the implications of the results and how these findings contribute to our current understanding of brain stimulation. Others studies using rTMS have demonstrated that it can modulate subcortical areas (https://doi.org/10.1016/j.crneur.2022.100033 ) and alter neurogenesis (ref 9 in the manuscript). Some discussion on how the current study fits in with these previous studies and its significance should be expanded on.

-It is interesting that the authors chose to include both male and female animals in their design. Noting the small sample size when subdividing the treatment groups by sex, the authors may want to include some quantitative details of this output (e.g. a table showing differences between sex in their treatment groups) as there seems to be a small difference between sex in Fig 2v iTBS which may be important to note for future studies. Nonetheless, some acknowledgement on the effects of sex on their results should be included in the discussion.

Minor comments

-The reference list seems to be duplicated and also slightly different between both lists. This should be amended.

-A reference should be supplied for the Allen mouse brain atlas (page 4).

-The enlarged images in the figures do not contain a scale bar (e.g. 1c, 1e, 2b-e, etc). Can a scale bar for these higher magnification images please be included.

-The y-axis for the bar graphs in the included figures should be a consistent scale (e.g. fig 2v).

6. PLOS authors have the option to publish the peer review history of their article (what does this mean? ). If published, this will include your full peer review and any attached files.

**Do you want your identity to be public for this peer review?**  For information about this choice, including consent withdrawal, please see our Privacy Policy .

Reviewer #1: **Yes: ** Alex Tang

Reviewer #2: No

Reviewer #3: **Yes: ** Jamie Beros

---

## [Author Response · Author response to Decision Letter 0]

5 Nov 2024

[These comments will read more clearly in the 'response to reviewers' file.]

Please find below our responses to the reviewers’ comments and suggestions, in bolded text. We have made many changes to the revised manuscript (highlighted in yellow in the marked-up version), which we feel significantly improves the quality of the report.

Reviewer #1: Gaertner et al have investigated the effect of two neurostimulation techniques on the survival or adult born striatal neurons. The study specifically studies the effect of the stimulation on neurons born prior to the stimulation regime. Although the study has some small sample sizes with some animals only have 1 neuron counted for dendritic spine density, it is well designed and well written. Publishing negative results is critical and informative and the authors should be congratulated on publishing these negative findings.

I only have minor corrections that should be easy to address.

Methods:

- please provide an estimate of the electric field induced or at least a dB/dT value for the rTMS stimulation. Presumably a biphasic sine wave was used?

Thank you for noting this important aspect in neuromodulation studies.

We used the Cool-40 Rat Coil (Parthoens, J., Verhaeghe, J., Servaes, S., Miranda, A., Stroobants, S., Staelens, S., 2016. Performance Characterization of an Actively Cooled Repetitive Transcranial Magnetic Stimulation Coil for the Rat. Neuromodulation: Technology at the Neural Interface 19, 459–468. https://doi.org/10.1111/ner.12387 )

The coil has a self induction of 9,0, a max. di/dt 200A/µsec at 100% amplitude, and a pulse width of 262µsec. The maximum stimulation output used was 32%. As described in the paper, and the literature on performance of coils, the di/dt follows a linear response, and therefore the di/dt is 64A/µsec at 32%

Alekseichuk and colleagues characterized the e-field modelling of the Cool-40 coil estimating the absolute value of Emax at 180 mV/mm. (Alekseichuk, I., Mantell, K., Shirinpour, S., Opitz, A., 2019. Comparative modeling of transcranial magnetic and electric stimulation in mouse, monkey, and human. Neuroimage 194, 136–148. https://doi.org/10.1016/j.neuroimage.2019.03.044 ).

Supplemental figure S5 from the above referenced paper.

We have added this information to the methods section of the paper.

- please check whether the duration listed (189 seconds) for iTBS is correct.

Good question. We looked into this and indeed it is correct. The reason why the duration appears different from what one would calculate based on the stated train durations etc is because each cycle is not exactly 10 sec as we stated but in fact is 9.82 sec.

- Was there a rationale for delivering rTMS every second day? rTMS is most commonly given daily for a period. An alternative protocol is fine to use, but a description of how that rationale was chosen would be useful.

Thank you for pointing out this difference between the schedule we use and the most frequent use in clinical practice.

We are reporting in this paper secondary analyses of a set of experiments where we wanted to understand cellular and mollecular mechanisms of action of both ECS and TMS. While ECT is delivered twice or thrice a week (never on consecutive days), rTMS is most often delivered daily. We opted for harmonizing the protocols for every other day, considering the above and the fact that rTMS has been also delivered in clinical settings every other day and showing similar outcomes to daily treatments (Galletly, C., Gill, S., Clarke, P., Burton, C., Fitzgerald, P.B., 2012. A randomized trial comparing repetitive transcranial magnetic stimulation given 3 days/week and 5 days/week for the treatment of major depression: is efficacy related to the duration of treatment or the number of treatments? Psychol. Med. 42, 981–988. https://doi.org/10.1017/S0033291711001760 ).

Nonetheless, the above is included in the manuscript for clarity.

- please provide the NA for the objectives used.

These details have been added.

- was there a reason 1-6 cells was counted per animal for dendritic spines? Counting 1 neuron vs 6 neurons is likely to have a large impact on the interpretation of the results.

Good question. The main reason is simply because there are variable numbers of newborn neurons in the striatum of adult mice. For this reason, we did not calculate animal-level averages, but instead include data from each neuron in the dataset. By using mixed-effects models we can therefore effectively include all of the neuronal data while at the same time account for variation that may arise from animal-level differences.

Discussion:

- It would be helpful to discuss the impact of the following on the results and their interpretation.

(1) the use of an rTMS coil that stimulates the entire brain and at least some of the spinal cord if not most of the peripheral nerves. This coil is known to activate neurons in the spinal cord of rats, and would therefore be reasonable to assume is not focal at all for the mouse brain.

Based on e-field modelling work shared above, it would appear the it is rather unlikely to have suprathreshold effects outside of the brain (see S5 image above).

(2) the use of a non-standard rTMS protocol. Potentially a lack of effect could be due to the spacing of the stimulation. While the results are interesting, noting that other rTMS protocols could still have potential is worth noting.

Agreed – this is a good point. In the revised manuscript we have raised this issue in the discussion.

Reviewer #2: The ms by Gaertner and colleagues describes the effects of ECT and rTMS on the survival and spine density of adult-born striatal neurons in mice. Both techniques did not alter striatal neurons survival and structural plasticity with the adopted stimulation patterns, which still needs to be pointed out.

Though of interest, the ms suffers from an unclear vocabulary, thus needing a strong language revision.

I have some major and few minor concerns.

Major points:

Introduction: Recent literature of rTMS effects on hippocampus, mPFC and M1 structural plasticity reported a stimulation-dependent modulation in mice; high-frequency rTMS increased dendritic plasticity in prefrontal and motor cortex, while 1 Hz stimulation induced structural plasticity in mature granule cells, as also dendritic complexity of newly generated neurons, though not increasing DG progenitor cells proliferation. These data shuld be discussed.

These are good points – we have some discussion about different effects of stimulation patterns on neural plasticity.

Methods: animals should be be better described in the initial paragraph.

Thanks for this advice. We have re-read the first paragraph of the methods to check for clarity.

Methods: Why did the average motor threshold was determined with a separate group of animals?

This was simply to avoid having to individually determine the motor threshold for each mouse on each day, since increased the time required to treat the mice would add unnecessary stress since mice had to be restrained.

Results: better to specify what the markers RFP, DARPP-32 and NeuN are used for.

This has been added.

Minor points:

Introduction: The sentence "rTMS and electroconvulsive shock (ECS), the animal model of ECT,..." is not clear at all. Please rephrase.

We have adjusted this sentence for clarity.

Introduction: The sentence "More recent studies indicate that neurogenesis in the ventral striatum is increased in response to chronic pain" needs more details.

We agree it would be nice to include more information about this interesting study but our main point was to highlight several studies that have shown evidence for adult neurogenesis in the striatum, which is otherwise a fairly rarely described phenomenon. This study otherwise has relatively little relationship to our study so we felt it would be distracting to add too many other details from it to the introduction.

Methods: The sentence "Sham-stimulated mice were restrained and treated identically with the exception that the stimulation intensity" is not at all clear!! Please rephrase.

This sentence has been rewritten.

Methods: specify the microscope and objective used for the acquisition of images

This has been changed.

Figure legend 3: modify "Each colored circle" with "each dot".

Reviewer #3: In their manuscript, Gaertner et al have investigated the effects of electroconvulsive shock (ECT) and 10Hz repetitive transcranial magnetic stimulation (rTMS) on the survival of adult born striatal neurons in the mouse brain. The author’s have used a transgenic mouse line to label adult-born neurons via tamoxifen injection, and beginning from 8 days later, applied ECT or iTBS rTMS for 10 sessions over 3 weeks. Neither ECT or rTMS induced changes in the amount of surviving adult-born striatal neurons or the density of dendritic spines. The manuscript is written well the findings are of interest to the field. My comments are detailed below:

Major comments

-The authors describe that they administered rTMS using a Cool-40 coil at 115% of motor threshold. Can the authors please provide detail on the estimated magnetic field or electric field at the surface of the cortex in their model. Additionally, there is evidence to suggest that this coil may lack focality and stimulate regions of the brain and spinal cord outside of the targeted region (https://doi.org/10.1111/ner.12387 ). Some acknowledgements of the non-focality of this coil should be included in the methods or discussion.

Thank you for the comment. We were not able to find definite evidence that Cool-40 coil stimulates areas of the spinal cord.

The paper referenced above does not substantiate such claims but rather raises the question based on the lack of difference in MEP latency in a set of experiments the authors do not provide data for or details as to how they conducted such experiment (Parthoens, J., Verhaeghe, J., Servaes, S., Miranda, A., Stroobants, S., Staelens, S., 2016. Performance Characterization of an Actively Cooled Repetitive Transcranial Magnetic Stimulation Coil for the Rat. Neuromodulation: Technology at the Neural Interface 19, 459–468. https://doi.org/10.1111/ner.12387 )

Page 465:

“Focality of the Electric Field Distribution

The induced E-field in the spherical head model has a ringshaped distribution as was expected for a circular TMS coil (17) and together with the 40 mm outer coil diameter suggests a rather unfocal stimulation of the rat cerebral cortex. This was confirmed by the lack of laterality in the MT determination. The average latencies of the hind limb MEPs in the present study (8.23 6 0.14 msec and 8.38 6 0.16 msec) were comparable to those previously described after more focal stimulation of the rat’s motor cortex using a figure-of-eight coil (8.76 6 0.29 msec) (27). However, using our described coil we found comparable latencies when placing the coil directly onto the spinal cord of a propofol-anesthetized rat (data not shown). This lack of focality and lack of differences in the latency times raises the question whether the evoked potentials measured in the rat’s limbs in studies using a circular coil might be, solely or in part, caused by stimulation of regions further down the corticospinal tract.”

The author’s speculation does not seem supported by modeling results by Alekseichuk and colleagues three years later in 2019 (Alekseichuk, I., Mantell, K., Shirinpour, S., Opitz, A., 2019. Comparative modeling of transcranial magnetic and electric stimulation in mouse, monkey, and human. Neuroimage 194, 136–148. https://doi.org/10.1016/j.neuroimage.2019.03.044 ).

-Were animals that were delivered ECT anaesthetised for the duration of the stimulation or was this conducted in non-anesthetised animals? If it was conducted in anesthetised animals, can details of the anaesthesia protocol please be detailed.

Good catch. Animals were anesthetized for the duration of the stimulation. We have clarified this in the methods.

-Can the authors please provide information on why the stimulation protocol was chosen. Why was stimulation delivered every second day? Also, why was tissue collected 3 days after the final stimulation day? Some discussion of how this stimulation protocol compares to other stimulation protocols and how the timing of tissue processing may have contributed to the results should be included.

See our response to a similar question about the stimulation protocol from Reviewer #1, which have addressed through changes to the text. We have also added a statement to the discussion which notes that neurostimulation could induce plasticity in new neurons at timepoints other than the ones we investigated.

-The discussion does not discuss the implications of the results and how these findings contribute to our current understanding of brain stimulation. Others studies using rTMS have demonstrated that it can modulate subcortical areas (https://doi.org/10.1016/j.crneur.2022.100033 ) and alter neurogenesis (ref 9 in the manuscript). Some discussion on how the current study fits in with these previous studies and its significance should be expanded on.

To the best of our knowledge our work is the first to compare and contrast the effects of ECS and rTMS side by side. Moreover, while many studies have investigated hippocampal neurogenesis, there is only one study of ECS effects on striatal neurogenesis, and no studies of TMS effects on striatal neurogenesis. Ourt study therefore offers translational insights about the different mechanism of action of these two modalities. In the revised manuscript we have added text at the beginning of the discussion to emphasize the unique aspects of our study.

-It is interesting that the authors chose to include both male and female animals in their design. Noting the small sample size when subdividing the treatment groups by sex, the authors may want to include some quantitative details of this output (e.g. a table showing differences between sex in their treatment groups) as there seems to be a small difference between sex in Fig 2v iTBS which may be important to note for future studies. Nonetheless, some acknowledgement on the effects of sex on their results should be included in the discussion.

This is a good point. We didn’t want to draw too much attention to sex given that we were underpowered to perform a proper analysis, but the reviewer indeed noticed a more than twofold difference between ECS-treated males and sham males. We have noted this in the results and also mention it in the discussion in light of the greated prevalence of PD in men than in women.

Minor comments

-The reference list seems to be duplicated and also slightly different between both lists. This should be amended.

Thanks for catching this; the duplicate reference list has been deleted.

-A reference should be supplied for the Allen mouse brain atlas (page 4).

This has been added.

-The enlarged images in the figures do not contain a scale bar (e.g. 1c, 1e, 2b-e, etc). Can a scale bar for these higher magnification images please be included.

This has been added.

-The y-axis for the bar graphs in the included figures should be a consistent scale (e.g. fig 2v).

This has been corrected.

---

## [Decision Letter · Decision Letter 1]

22 Nov 2024

PONE-D-24-35662R1Electroconvulsive shock and transcranial magnetic stimulation do not alter the survival or spine density of adult-born striatal neurons

PLOS ONE

Dear Dr. Snyder,

Thank you for resubmitting your work to PLOS ONE. Please make the corrections posed by the Reviewers so I can render a decision on this manuscript.

**Comments to the Author**

1. If the authors have adequately addressed your comments raised in a previous round of review and you feel that this manuscript is now acceptable for publication, you may indicate that here to bypass the “Comments to the Author” section, enter your conflict of interest statement in the “Confidential to Editor” section, and submit your "Accept" recommendation.

Reviewer #1: (No Response)

Reviewer #2: All comments have been addressed

Reviewer #3: (No Response)

2. Is the manuscript technically sound, and do the data support the conclusions?

Reviewer #1: Partly

Reviewer #2: Yes

Reviewer #3: Yes

3. Has the statistical analysis been performed appropriately and rigorously? 

Reviewer #1: Yes

Reviewer #2: Yes

Reviewer #3: Yes

4. Have the authors made all data underlying the findings in their manuscript fully available?

Reviewer #1: Yes

Reviewer #2: Yes

Reviewer #3: (No Response)

5. Is the manuscript presented in an intelligible fashion and written in standard English?

Reviewer #1: Yes

Reviewer #2: Yes

Reviewer #3: Yes

6. Review Comments to the Author

Reviewer #1: -The authors state the induced e field for their coil was 180V/m as per the modelling results from the Opitz lab. However, that simulation was done with an input of 100uA/us. This is not the input used in the current study, so the estimate of 180V/m is not correct.

- The authors maintain that the coil used is focal however I still strongly disagree. The coil is larger than the mouse head, with the windings sitting over the spinal cord unless the coil was offset such that the coil was overhanging at the front of the head to avoid the spinal cord. This would however distort the e-field. Moreover, the e-field is not restricted to a small portion of the cortex, it stimulates essentially the whole brain as well. Therefore, I maintain that the authors need to acknowledge the lack of focality in their discussion and make some mention of how this would impact their results.

- I accept the comment about low numbers being due to low number of adult born cells. This should be mentioned in the discussion as a limitation.

Reviewer #2: The Authors cleared all my concerns. I only have one minor issue: please modify "noninvasive" with "non-invasive"

Reviewer #3: I would like to thank the authors for considering and replying to the reviewer comments. Most of the comments are adequately addressed and the manuscript with the new inclusions read well. I have minor comments below that I believe still need to be addressed:

-The authors have replied to the reviewer’s comment and refute the claim that the coil used in the study does not stimulate the spinal cord as stated in the original coil validation paper and cite an additional modelling paper. The manuscript and figure that the author’s provided to substantiate their claims show that other areas across the surface of the brain (e.g. lateral cortex, cerebellum and olfactory bulbs) display an increased electric field with this coil, showing that areas outside of the targeted area can be affected. My comment was intended to consider how the focality of the applied stimulation (what/how many areas of the brain are being stimulated by the coil) led to/did not lead to the outcomes observed in the study rather than if it specifically stimulates the spinal cord. I still believe that it is important for the authors to at least acknowledge the claim in the original validation paper that the coil may stimulate areas outside of the targeted area within their manuscript. The authors can discuss/include their reasoning refuting the focality claim there if they wish.

-For the new sentence “While we did not have sufficient group sizes to investigate sex differences, it is worth noting that ECS-treated males had more than twice as many new neurons than their sham counterparts, but the two groups of females were virtually identical”, can the authors please include the mean value from these groups (treatment vs sham) in this sentence to aid in the comparison.

7. PLOS authors have the option to publish the peer review history of their article (what does this mean? ). If published, this will include your full peer review and any attached files.

**Do you want your identity to be public for this peer review?**  For information about this choice, including consent withdrawal, please see our Privacy Policy .

Reviewer #1: **Yes: ** Alex Tang

Reviewer #2: No

Reviewer #3: **Yes: ** Jamie Beros

We look forward to receiving your revised manuscript.

Kind regards,

Stephen D. Ginsberg, Ph.D.

Section Editor

PLOS ONE
---

## [Author Response · Author response to Decision Letter 1]

13 DEC 2024

These same comments are also provided (in a more readable fashion) in the response to reviewers file.

Please find below our responses to the reviewers’ comments and suggestions, in bolded text.

Reviewer #1: -The authors state the induced e field for their coil was 180V/m as per the modelling results from the Opitz lab. However, that simulation was done with an input of 100uA/us. This is not the input used in the current study, so the estimate of 180V/m is not correct.

- The authors maintain that the coil used is focal however I still strongly disagree. The coil is larger than the mouse head, with the windings sitting over the spinal cord unless the coil was offset such that the coil was overhanging at the front of the head to avoid the spinal cord. This would however distort the e-field. Moreover, the e-field is not restricted to a small portion of the cortex, it stimulates essentially the whole brain as well. Therefore, I maintain that the authors need to acknowledge the lack of focality in their discussion and make some mention of how this would impact their results.

This has been acknowledged and in the revised discussion we discuss the potential consequences.

- I accept the comment about low numbers being due to low number of adult born cells. This should be mentioned in the discussion as a limitation.

This limitation has been added to the discussion.

Reviewer #2: The Authors cleared all my concerns. I only have one minor issue: please modify "noninvasive" with "non-invasive"

This has been corrected.

Reviewer #3: I would like to thank the authors for considering and replying to the reviewer comments. Most of the comments are adequately addressed and the manuscript with the new inclusions read well. I have minor comments below that I believe still need to be addressed:

-The authors have replied to the reviewer’s comment and refute the claim that the coil used in the study does not stimulate the spinal cord as stated in the original coil validation paper and cite an additional modelling paper. The manuscript and figure that the author’s provided to substantiate their claims show that other areas across the surface of the brain (e.g. lateral cortex, cerebellum and olfactory bulbs) display an increased electric field with this coil, showing that areas outside of the targeted area can be affected. My comment was intended to consider how the focality of the applied stimulation (what/how many areas of the brain are being stimulated by the coil) led to/did not lead to the outcomes observed in the study rather than if it specifically stimulates the spinal cord. I still believe that it is important for the authors to at least acknowledge the claim in the original validation paper that the coil may stimulate areas outside of the targeted area within their manuscript. The authors can discuss/include their reasoning refuting the focality claim there if they wish.

As also requested by Reviewer 1, focality limitations and consequences have been added to the discussion.

-For the new sentence “While we did not have sufficient group sizes to investigate sex differences, it is worth noting that ECS-treated males had more than twice as many new neurons than their sham counterparts, but the two groups of females were virtually identical”, can the authors please include the mean value from these groups (treatment vs sham) in this sentence to aid in the comparison.

This has been added.

---

## [Editor Report · Decision Letter 2]

17 Dec 2024

Electroconvulsive shock and transcranial magnetic stimulation do not alter the survival or spine density of adult-born striatal neurons

PONE-D-24-35662R2

Dear Dr. Snyder,

We’re pleased to inform you that your manuscript has been judged scientifically suitable for publication and will be formally accepted for publication once it meets all outstanding technical requirements.

Kind regards,

Stephen D. Ginsberg, Ph.D.

Section Editor

PLOS ONE